

# Alterations in SARS-CoV-2 Omicron and Delta peptides presentation by HLA molecules

Stepan Nersisyan[1,2,3,4], Anton Zhiyanov[2], Maria Zakharova[2], Irina Ishina[2], Inna Kurbatskaia[2], Azad Mamedov[2], Alexei Galatenko[1,5], Maxim Shkurnikov[1], Alexander Gabibov[2] and Alexander Tonevitsky[1,2,6]

[1] Faculty of Biology and Biotechnology, HSE University, Moscow, Russia
[2] Shemyakin-Ovchinnikov Institute of Bioorganic Chemistry, Russian Academy of Sciences, Moscow, Russia
[3] Institute of Molecular Biology, The National Academy of Sciences of the Republic of Armenia, Yerevan, Armenia
[4] Armenian Bioinformatics Institute (ABI), Yerevan, Armenia
[5] Faculty of Mechanics and Mathematics, Lomonosov Moscow State University, Moscow, Russia
[6] Art Photonics GmbH, Berlin, Germany

Corresponding author
Maxim Shkurnikov,
mshkurnikov@gmail.com

## ABSTRACT

The T-cell immune response is a major determinant of effective SARS-CoV-2 clearance. Here, using the recently developed T-CoV bioinformatics pipeline (https://t-cov.hse.ru) we analyzed the peculiarities of the viral peptide presentation for the Omicron, Delta and Wuhan variants of SARS-CoV-2. First, we showed the absence of significant differences in the presentation of SARS-CoV-2-derived peptides by the most frequent HLA class I/II alleles and the corresponding HLA haplotypes. Then, the analysis was limited to the set of peptides originating from the Spike proteins of the considered SARS-CoV-2 variants. The major finding was the destructive effect of the Omicron mutations on PINLVRDLPQGFSAL peptide, which was the only tight binder from the Spike protein for HLA-DRB1*03:01 allele and some associated haplotypes. Specifically, we predicted a dramatical decline in binding affinity of HLA-DRB1*03:01 and this peptide both because of the Omicron BA.1 mutations (N211 deletion, L212I substitution and EPE 212-214 insertion) and the Omicron BA.2 mutations (V213G substitution). The computational prediction was experimentally validated by ELISA with the use of corresponding thioredoxin-fused peptides and recombinant HLA-DR molecules. Another finding was the significant reduction in the number of tightly binding Spike peptides for HLA-B*07:02 HLA class I allele (both for Omicron and Delta variants). Overall, the majority of HLA alleles and haplotypes was not significantly affected by the mutations, suggesting the maintenance of effective T-cell immunity against the Omicron and Delta variants. Finally, we introduced the Omicron variant to T-CoV portal and added the functionality of haplotype-level analysis to it.

## INTRODUCTION

T-cell immune response plays a pivotal role in the pathogenesis of COVID-19 (*Sekine et al., 2020*; *Nelde et al., 2021*; *Shkurnikov et al., 2021*). Cytotoxic (CD8) T-cells become activated through recognition of viral peptides presented by HLA class I molecules on the surface of antigen-presenting cells (APCs). The same mechanism based on recognition of HLA-I/peptide complex is further used to identify and destroy infected cells. Unlike cytotoxic T-cells, helper (CD4) T-cells become activated through interaction between their T-cell receptors (TCR) and viral peptides presented by HLA class II proteins. One of the main effector functions of helper T-cells consists in delivering the second activation signal to B-cells, which is necessary for the initiation of antibody production (*Kumar, Connors & Farber, 2018*).

HLA genetics were extensively studied in the context of COVID-19 susceptibility and severity (*Augusto & Hollenbach, 2022*). A number of manuscripts reported several risk and protective alleles for COVID-19; for example, HLA-C*04:01 carrier state was associated with severe disease course in Armenia (*Hovhannisyan et al., 2022*), Germany, Spain, Switzerland and the United States (*Weiner et al., 2021*). Other studied showed the association between the number of viral peptides presented by an individual's HLA molecules set and COVID-19 severity (*Iturrieta-Zuazo et al., 2020*; *Shkurnikov et al., 2021*). Nevertheless, the existing links between HLA genetics and COVID-19 pathogenesis are highly population-specific (*Augusto & Hollenbach, 2022*). It should be also added that some HLA alleles/genotypes were shown to be associated with susceptibility to other respiratory diseases such as Influenza A (*Falfán-Valencia et al., 2018*).

Recently emerged SARS-CoV-2 variants effectively escape neutralization by antibodies directed to the Spike protein of the base Wuhan variant. According to Nextstrain project, at the beginning of April 2022 most of new COVID-19 cases were driven by Omicron and Delta variants. While replication and antibody-mediated neutralization of these variants were studied extensively (*Shiehzadegan et al., 2021*; *Cameroni et al., 2021*; *Zhao et al., 2022*), the role of T-cell immune response and possibility of T-cell immunity evasion are to be uncovered. In several recent studies preservation of robust T-cell immunity against the Omicron variant was suggested (*GeurtsvanKessel et al., 2021*; *Keeton et al., 2021*; *May et al., 2021*; *Liu et al., 2022*; *Mazzoni et al., 2022*). However, a more elaborative analysis should be conducted to address the population-level diversity of HLA molecules.

We recently developed T-cell COVID-19 Atlas (T-CoV)—the computational pipeline and web portal for evaluation of impact of SARS-CoV-2 mutations on HLA-peptide interactions (*Nersisyan et al., 2022*). Here we used T-CoV to compare viral peptide presentation for the three variants: Wuhan, Delta and Omicron. Since the major part of the existing vaccines are based on the Spike protein of the reference variant (*Kyriakidis et al., 2021*), the comparisons were separately conducted for the whole virus and Spike protein peptidomes. At the beginning, the analysis was performed on a single allele-level; 64 HLA class I (HLA-A, HLA-B, HLA-C) and 105 HLA class II (HLA-DR, HLA-DQ, HLA-DP) abundant alleles were screened. Then, the most relevant findings (differential HLA-peptide interactions) were validated experimentally with ELISA. Next, the considered alleles were

combined into the theoretical library of all possible haplotypes, and the peptide presentation analysis was conducted at the level of haplotypes. Finally, The Allele Frequency Net Database was utilized to highlight the most frequent haplotypes with altered Omicron/Delta peptide presentation (*Gonzalez-Galarza et al., 2020*). The workflow of the study is presented in Fig. 1.

## MATERIALS AND METHODS

### Bioinformatics analysis of HLA/peptide interactions

Protein sequences of SARS-CoV-2 variants were obtained from GISAID (*Elbe & Buckland-Merrett, 2017*):

- EPI_ISL_402125 (Wuhan);
- EPI_ISL_1663516 (Delta, B.1.617.2);
- EPI_ISL_6699752 (Omicron BA.1, B.1.1.529);
- EPI_ISL_9884589 (Omicron BA.2, B.1.1.529).

T-CoV pipeline was executed for the analysis of HLA/peptide interactions (*Nersisyan et al., 2022*). Briefly, binding affinities of all viral peptides and 169 frequent HLA class I/II molecules were predicted with NetMHCpan 4.1 and NetMHCIIpan 4.0 (*Reynisson et al., 2020*). Then, predicted affinities were compared between the reference Wuhan variant and Omicron/Delta. HLA/peptide pairs whose affinities were altered by at least two folds were used in the downstream analysis.

### HLA-DRB1*03:01/peptide binding experiments

HLA-DR/peptide binding experiments were conducted as we previously described (*Mamedov et al., 2020*). Briefly, the genetic constructions for recombinant HLA-DR α and β (HLA-DRB1*03:01) chains with CLIP expression in HEK293F suspension cells (ATCC, Manassas, VA, USA) were created based on pFUSE vector encoding constant fragment of human immunoglobulin heavy chain (Fc). CLIP (PVSKMRMATPLLMQA) was covalently attached with the linker with a thrombin site at the N-terminus of β chain.

Seven thioredoxin-fused 15-mer peptides were constructed to experimentally validate NetMHCIIpan-predicted differences in HLA-DRB1*03:01 binding because of the Omicron mutations:

- PINLVRDLPQGFSAL (the reference Wuhan/Delta peptide);
- TPIIVREPEDLPQGF, IIVREPEDLPQGFSA, VREPEDLPQGFSALE, EPEDLPQGFSALEPL (four Omicron BA.1 peptides with different shifts compared to the reference sequence);
- PINLGRDLPQGFSAL, KHTPINLGRDLPQGF (two Omicron BA.2 peptides with different shifts compared to the reference sequence).

The substrate construct, carrying only thioredoxin with the linker, was used as a negative control. Thioredoxin-fused peptides were chemically biotinylated with EZ-Link Sulfo-NHS-LC-biotin (Thermo Fisher Scientific, Waltham, MA, USA).

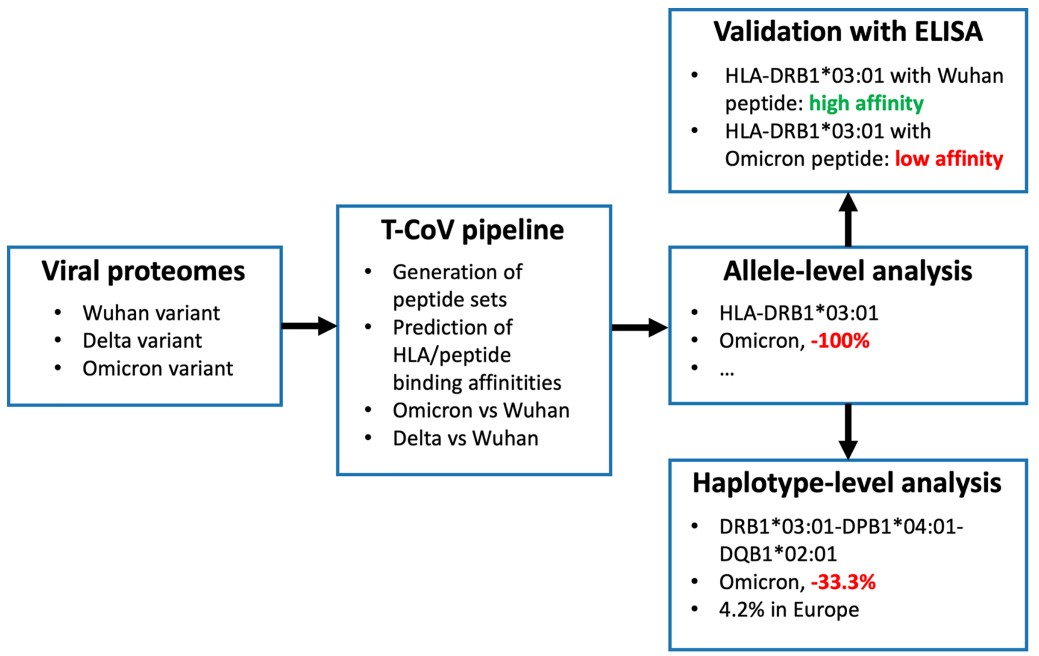

**Figure 1 Workflow of the analysis.** Three panels on the right illustrate the results for HLA-DRB1*03:01 allele and DRB1*03:01-DPB1*04:01-DQB1*02:01 haplotype.

Biotinylated and thioredoxin-fused peptides (750 nM) were incubated overnight at 37 °C in PBS in 50 μl with HLA-DR (HLA-DRB1*03:01, 150 nM). L243 mAb (5 μg/ml in PBS) were immobilized on the 96-well flat-bottom Maxisorp ELISA plate (Nunc, Waltham, MA, USA) overnight at 4 °C. Wells were blocked then with 1% milk in PBS for 1 h at 37 °C with shaking. Afterwards DR/peptide complexes were captured with immobilized L243 mAb for 1 h at 37 °C. HLA-DR-bound biotinylated peptide was quantitated with streptavidin-peroxidase (50 μl in PBST with dilution 1:5,000) (Abcam, Cambridge, UK) by incubation for 1 h at 37 °C, using 3,3′,5,5′-tetramethylbenzidine (50 μl) as a substrate for 5 min and stopping with 10% phosphoric acid (50 μl). Between all stages the wells were washed three times with PBST. Absorption of signals was measured at 450 nm using Varioscan plate reader (Thermo Fisher Scientific, Waltham, MA, USA).

## HLA allele and haplotype frequency analysis

HLA allele and haplotype frequencies were downloaded from Allele Frequency Net Database (*Gonzalez-Galarza et al., 2020*) for the following regions: Europe, North America, North-East Asia, South Asia, South and Central America, South-East Asia and Western Asia. For HLA class I haplotypes three genes were selected: HLA-A, HLA-B, HLA-C. For HLA class II three haplotype pools were analyzed separately:

- HLA-DRB1, HLA-DQB1, HLA-DPB1;
- HLA-DQA1, HLA-DQB1;
- HLA-DPA1, HLA-DPB1.
In addition to the worldwide frequency data, we used our previously described dataset of 428 volunteers to assess the HLA alleles distribution in Moscow, Russia (*Shkurnikov et al., 2021*). Briefly, HLA-A, HLA-B, HLA-C, HLA-DRB1, HLA-DQB1 genes were sequenced with the MiSeq platform (Illumina, San Diego, CA, USA) using reagent kit HLA-Expert (DNA-Technology LLC, Moscow, Russia). Frequencies of HLA-A/B/C and HLA-DRB1/DQB1 haplotypes were inferred with Hapl-o-Mat v1.1; the expectation-maximization algorithm with default settings was used (*Schäfer, Schmidt & Sauter, 2017*).

# RESULTS

## Design of the allele-level analysis

With the use of T-CoV database we obtained the lists of viral peptides which were predicted to be tight binders (affinity ≤ 50 nM) for the worldwide prevalent HLA class I and class II alleles. The analysis was conducted for three SARS-CoV-2 variants: Wuhan, Delta and Omicron (BA.1 and BA.2). Since the immune system of many individuals was exposed only to Spike protein of the reference Wuhan variant (Spike protein-based vaccination), we used two peptide pools: the whole peptidome of the virus and the peptidome of the Spike protein.

First, we performed individual allele-level analysis. Given the numbers of tightly binding peptides for the fixed HLA allele and three SARS-CoV-2 variants, we calculated the differences in the numbers of tight binders of the Delta and Omicron variants relative to the Wuhan virus (Table S1). For example, −100% difference indicated complete vanishing of all tightly binding Wuhan virus peptides, while 100% increase indicated doubling the number of peptides. Note that denominators included the number of tight binders in all viral proteins or solely Spike protein depending on the considered pool of peptides.

## Ten HLA class I alleles had altered presentation of peptides from the Spike proteins of the Omicron and Delta variants

Only one out of 64 common HLA class I alleles was significantly affected at the level of the whole virus: a single peptide from both Delta and Omicron (BA.1, BA.2) variants (FP**L**TSFGPL) originating from the NSP12 protein became tight binder for HLA-B*35:03 because of P323L substitution (50% relative increase in the number of tight binders).

Much more alleles (ten) demonstrated significantly altered peptide presentation (≤−25% or ≥25%) during the analysis of the Spike protein peptides (Fig. 2). Eight alleles were found in the context of the Omicron variant, and only three alleles showed differential presentation of the Delta peptides (HLA-B*07:02 allele was marked for both variants). The results were also skewed in the direction of enhanced peptide presentation (7/10 alleles).

Three alleles, HLA-B*07:02, HLA-B*27:05 and HLA-A*32:01, had more than 50% difference in the number of presented Omicron or Delta peptides, while seven other alleles (HLA-C*01:02, HLA-B*44:03, HLA-B*18:01, HLA-A*23:01, HLA-A*30:01, HLA-B*08:01, HLA-C*15:02) showed weaker difference in the peptide presentation. The impact of the Delta and Omicron (BA.1, BA.2) mutations on the peptide presentation was the same for

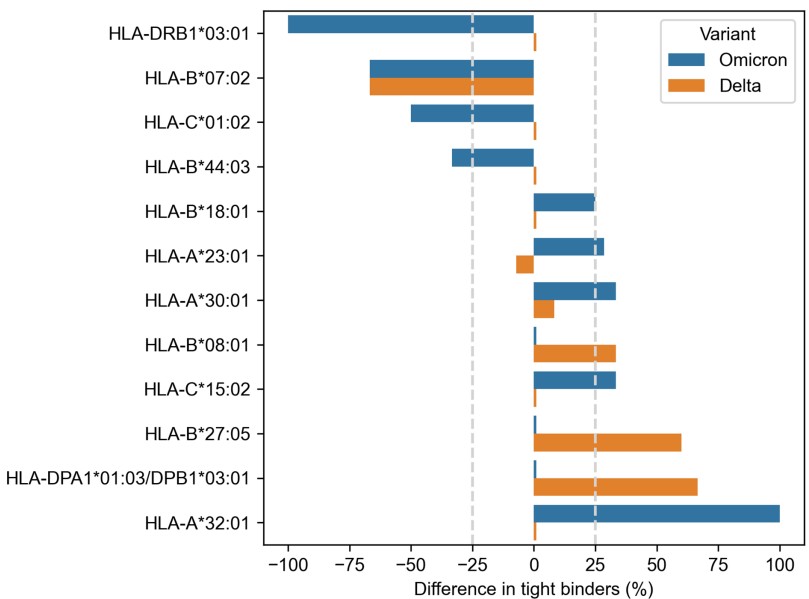

**Figure 2 Differences in viral peptide presentation for the Delta and Omicron variants.** The differences in the number of tight binders were normalized by the number of tight binders in the Wuhan variant.

HLA-B*07:02 allele: S**P**RRARSVA and three surrounding peptides lost their binding affinity because of P681H mutation. Next, L452R substitution in the Delta variant led to the emergence of three novel tight binders (Y**R**YRLFRK, Y**R**YRLFRKSNL, Y**R**YRLFRKSNLK) for HLA-B*27:05. Finally, two tight binders for HLA-A*32:01 appeared because of the mutations in the Omicron variant:

- Omicron BA.1: VLYN**LAP**FF (S371L, S373P, S375F) and **R**SY**S**F**R**PT**Y** (Q493R, G496S, Q498R, N501Y);
- Omicron BA.2: VLYN**FAP**FF (S371F, S373P, S375F) and **R**SYGF**R**PT**Y** (Q493R, Q498R, N501Y).

## HLA-DRB1*03:01 lost all tight binders from the Spike protein due to the mutations in the Omicron variant

Not a single HLA class II allele passed 25% difference threshold when the analysis was performed at the level of the whole virus peptidome. In the case of the Spike protein, two alleles were highlighted: HLA-DRB1*03:01 for the Omicron variant and HLA-DPA1*01:03/DPB1*03:01 for the Delta variant.

The striking observation consisted in fact that all predicted tight binders from the Spike protein of the Wuhan and Delta variants for HLA-DRB1*03:01 allele vanished because of the Omicron mutations. There were several tightly binding peptides for HLA-DRB1*03:01 centered around PINLVRDLPQGFSAL peptide with the nine amino acid core LVRDLPQGF (binding affinity 27 nM). The Omicron BA.1 variant had two mutations within the peptide core (L212I substitution, EPE 212-214 insertion) and N211 deletion in

the peptide flank (Fig. 3A). According to the predictions obtained by NetMHCIIpan, all corresponding Omicron peptides became non-binders for HLA-DRB1*03:01 allele (predicted affinity > 5,000 nM). Surprisingly, the Omicron BA.2 variant had different mutation in the core of the considered peptide (V213G substitution, Fig. 3B), which decreased HLA-DRB1*03:01 binding affinity by five folds according to NetMHCIIpan.

In order to experimentally validate the computational predictions, biotinylated and thioredoxin-fused reference PINLVRDLPQGFSAL peptide from the Wuhan/Delta Spike proteins, as well as six matching Omicron BA.1 and BA.2 peptides were constructed. Then, the constructed peptides and recombinant HLA-DR molecules (HLA-DRB1*03:01) were incubated together overnight, and DR/peptide complexes were captured on the ELISA plate. In full accordance with the bioinformatics predictions, the reference peptide bound HLA-DRB1*03:01, while the corresponding Omicron BA.1 peptides were not forming complexes with the same HLA-DR receptors and binding efficiency of the Omicron BA.2 peptides markedly reduced (Fig. 3C, Table S2).

The importance of the finding is emphasized by the high frequency of HLA-DRB1*03:01 allele in some regions, including 8.9% in Europe and 10.0% in Moscow, Russia. As we have already mentioned, presentation of the Omicron peptides by HLA-DRB1*03:01 was significantly different from the Wuhan variant only for the Spike protein, since there were several conserved tight binders in other proteins (N, NSP2, NSP3, NSP4, NSP5, NSP8, NSP12, NSP13, NSP14). Thus, the reduced presentation efficiency of HLA-DRB1*03:01 could possibly affect only individuals vaccinated with Spike protein-based vaccines.

There was another HLA class II allele with differential peptide presentation: HLA-DPA1*01:03/DPB1*03:01. Specifically, P681R mutation in the Delta variant slightly strengthened binding affinity of two peptides to the mentioned HLA-DP molecule: YQTQTNSPRRARSVASQSII (84 nM to 40 nM) and QTQTNSPRRARSVASQSIIA (66 nM to 33 nM).

## The Omicron and Delta mutations altered peptide presentation efficiency for several HLA class I and II haplotypes

Individual allele-level analysis allowed us to find HLA alleles with significantly altered peptide presentation. At the same time, each individual carries multiple alleles at once, including two parental alleles of HLA class I genes (HLA-A, HLA-B, HLA-C) and HLA class II genes (HLA-DRB1, HLA-DQA1, HLA-DQB1, HLA-DPA1, HLA-DPB1). All these genes are closely linked, and a whole HLA haplotype is inherited from each parent (Choo, 2007). Given that, a single "weakened" allele in a set of "strong" alleles will not affect much total peptide presentation by individual's HLA molecules set. To assess whether HLA haplotypes with significantly different presentation of Omicron/Delta peptidomes exist, two sets of haplotypes were analyzed:

1. Theoretical library of all possible haplotypes composed of HLA alleles under consideration.
2. Library of the most frequent HLA haplotypes in several populations.

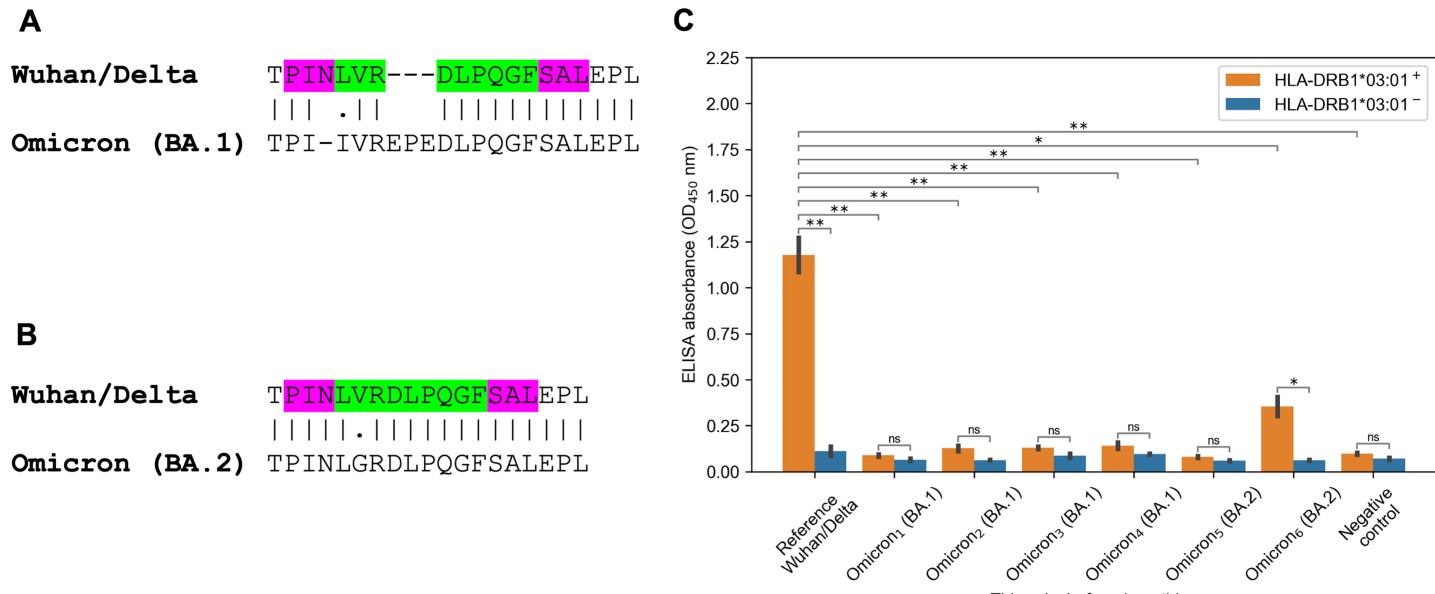

**Figure 3** **Mutations in the Omicron variant led to vanishing of HLA-DRB1\*03:01 tight binders.** (A and B) The region of consideration. Green color stands for HLA-DRB1\*03:01 9-mer binding core, purple color stands for the peptide flanking residues. (C) The results of ELISA analysis. $Omicron_1$, $Omicron_2$, $Omicron_3$, $Omicron_4$, $Omicron_5$, $Omicron_6$ stand for TPIIVREPEDLPQGF, IIVREPEDLPQGFSA, VREPEDLPQGFSALE, EPEDLPQGFSALEPL, PINLGRDLPQGFSAL and KHTPINLGRDLPQGF peptides. Negative control stands for the substrate construct, carrying only thioredoxin with the linker. \*\*: $p < 0.01$, \*: $p < 0.05$, ns: $p > 0.05$ (Student's $t$-test).

We also updated our T-CoV portal to allow users to perform the analysis with their set of HLA alleles (Fig. 4, https://t-cov.hse.ru/haplotypes).

For HLA class II we considered frequencies of HLA-DRB1/DPB1/DQB1 haplotypes since HLA-DPA1 and HLA-DQA1 genotyping was not performed in the majority of the available datasets. Because each HLA-DPB1 and HLA-DQB1 allele is closely linked with only few alpha chain variants, we then manually adjusted the results for HLA-DPA1/DPB1 and HLA-DQA1/DQB1 links. Namely, 1–3 possible alleles encoding alpha chains were associated with each HLA-DP and HLA-DQ beta chains.

Only 4 out of 9,576 theoretically possible HLA class I haplotypes had significantly enhanced Omicron/Delta peptide presentation at the level of the whole SARS-CoV-2 proteome (Table S3). As expected, all these genotypes contained HLA-B\*35:03 allele, which was the only selected entry in the individual allele analysis. The single haplotype (A\*25:01-B\*35:03-C\*04:01) was marked as frequent in Europe (frequency = 0.02%). Remarkably higher number of haplotypes (659 out of 9,576) were significantly affected at the level of the Spike protein. For the Delta variant, approximately equal numbers of haplotypes showed reduced (91 haplotypes) and enhanced (81 haplotypes) peptide presentation (Table S3). The situation was highly biased towards more effective presentation in the Omicron case: 453 and 100 haplotypes with the significant increase and decrease in numbers of tight binders, respectively.

Less than 10% of the identified haplotypes (54 out of 659) were present in the list of the most frequent HLA haplotypes. A\*24:02-B\*07:02-C\*07:02 haplotype had the highest frequency: 1.2% in Europe, 2.7% in North America, 4% in North-East Asia, 1.6% in South

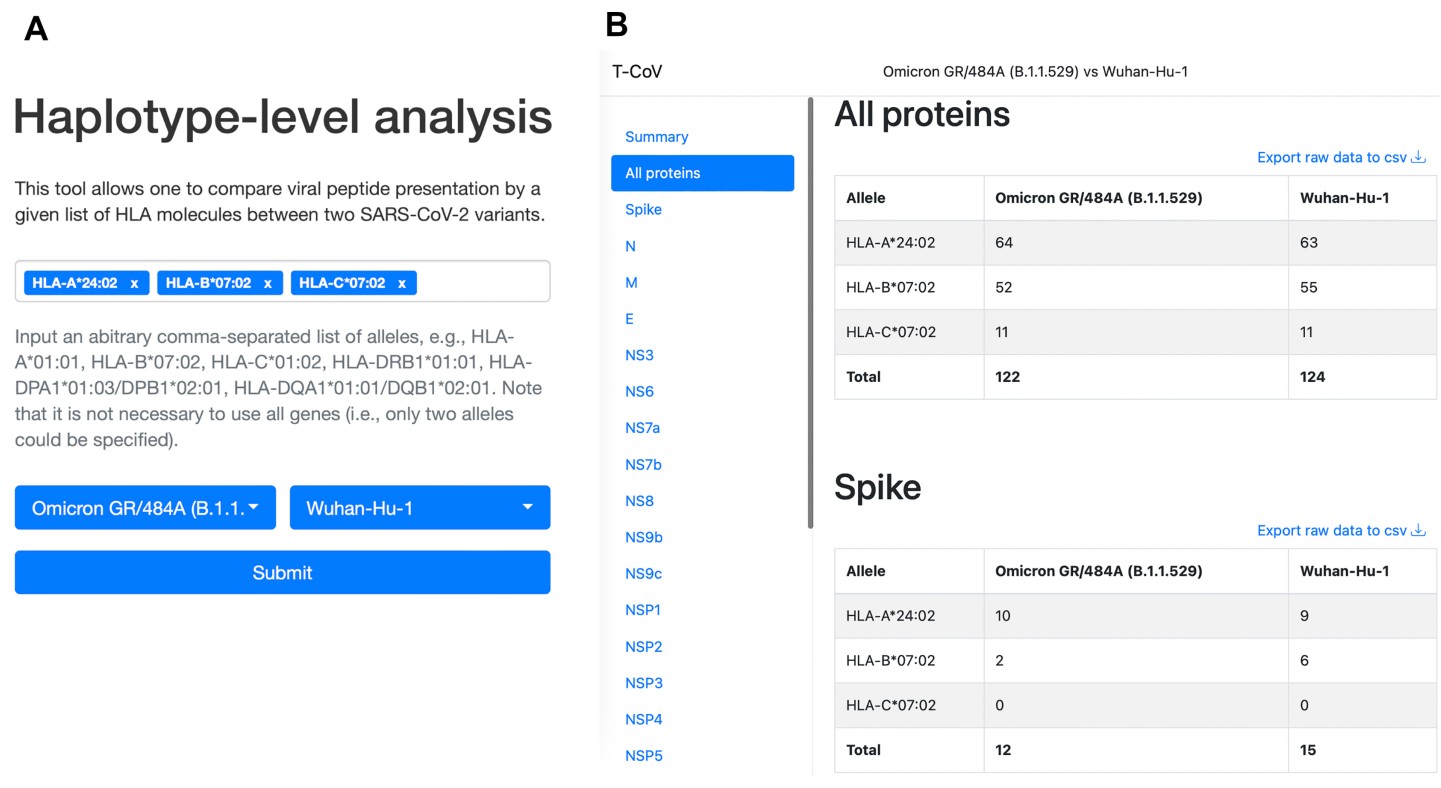

**Figure 4** **The web interface of T-CoV new tool.** (A) User can select HLA class I/II alleles and SARS-CoV-2 variants. (B) The results of analysis are grouped by viral proteins and entered alleles.

Asia and 1.4% in Moscow, Russia. Efficiency of viral peptide presentation by this set of HLA molecules was reduced by 20% and 33.3% for the Omicron and Delta variants, respectively.

Significant alterations in peptide presentation for HLA class II haplotypes were found only for the Spike protein case (Table S4). Surprisingly, these differences were completely opposite for the Delta and Omicron variants: 320 haplotypes were associated with enhanced presentation of Delta peptides, while 951 haplotypes were associated with reduced Omicron peptide presentation. Consistently with the individual allele-level analysis, all identified haplotypes contained HLA-DPA1*01:03/DPB1*03:01 allele for the Delta variant and HLA-DRB1*03:01 allele for the Omicron variant.

There were only four frequent HLA class II haplotypes affected by the Delta mutations and six haplotypes for the Omicron variant (Table S4). From them, DRB1*03:01-DPB1*04:01-DQB1*02:01 had especially high worldwide frequency: 4.2% in Europe, 10% in Western Asia and 10% in Moscow, Russia. Based on our computational analysis, the number of tight binding SARS-CoV-2 peptides for this haplotype is reduced by 33.3% in the Omicron variant compared to the reference Wuhan virus.

## DISCUSSION

In this manuscript we compared peptide presentation profiles for the Omicron, Delta and Wuhan SARS-CoV-2 variants. First, the analysis was performed at the level of individual

alleles. Only one allele with significantly altered peptide presentation was identified: HLA-B*35:03 had one more Omicron/Delta tight binding peptide in NSP12 protein. Thus, we predicted the complete absence of T-cell immunity evasion at the level of the whole virus. This agrees with the recent findings by Xiao and Qiu with co-authors: while several peptides lost binding ability to HLA-A*02 because of mutations, overall levels of T-cell immunity were not strongly decreased (*Qiu et al., 2021*; *Xiao et al., 2022*). When the analysis was limited to the Spike protein, remarkably higher number of alleles was identified: eight alleles showed enhanced ability of Omicron/Delta peptide presentation, while four remaining alleles showed significant decrease in the number of tight binders.

HLA-DRB1*03:01 molecule had the highest escape rate for the Omicron peptides: the only predicted tight binder from the Wuhan variant for this allele (PINLVRDLPQGFSAL) lost its binding affinity because of several mutations: N211 deletion, L212I substitution and EPE 212-214 insertion for the Omicron BA.1 variant and V213G substitution for the Omicron BA.2 variant. Aside from the high predicted binding affinity, CD4 T-cell immunogenicity of the mentioned peptide was previously validated in two experimental reports (*Keller et al., 2020*; *Verhagen et al., 2021*). The set of tight binding peptides for another allele, HLA-B*07:02, was also exhausted because of the Spike P681R mutation which was present both in the Omicron and Delta variants. In concordance with our findings, Hamelin with co-authors showed that epitopes associated with B07 supertype were likely to escape CD8 T-cell immunity during the first year of the pandemic (*Hamelin et al., 2021*).

Given the results of the allele-level analysis, we constructed theoretical libraries of HLA class I and II haplotypes composed of the considered alleles. Few haplotypes which included alleles with altered presentation of mutant peptides also showed significant differential peptide presentation. One of the identified HLA class II haplotypes had especially high worldwide frequency, including Europe, Western Asia and Moscow, Russia. Namely, the number of highly affine peptides for DRB1*03:01-DPB1*04:01-DQB1*02:01 was reduced by 33.3% in the Omicron variant compared both to the Wuhan and Delta. Individuals carrying this haplotype could possibly develop impaired CD4 T-cell response to the Omicron variant following Wuhan Spike protein-based vaccination, which would consequently imply impaired antibody response. Nevertheless, the overwhelming majority of haplotypes were not associated with significantly reduced Omicron/Delta peptide presentation, which fully agrees with the recently conducted experiments on small patient cohorts (*GeurtsvanKessel et al., 2021*; *Keeton et al., 2021*; *May et al., 2021*; *Liu et al., 2022*; *Mazzoni et al., 2022*).

Another important dimension in the HLA/COVID-19 research is the regulation of HLA genes expression during SARS-CoV-2 infection. In a recent report, *Zhang et al. (2022)* found a significant increase of HLA-B*18:01/B*44:03 allelic fold change in A549 cells infected with SARS-CoV-2. The possibility of altered regulation of HLA expression by new SARS-CoV-2 variants can deepen the current understanding of the mechanisms underlying the immune response evasion.

## CONCLUSIONS

The high diversity of HLA alleles and haplotypes coupled with the specificity of peptide presentation strongly limits the potential of T-cell immune response evasion of SARS-CoV-2. In this manuscript, we identified several HLA class I and II alleles with impaired presentation of peptides originating from the Spike protein of the Omicron and Delta variants. The strongest effect was observed for the HLA-DRB1*03:01 allele, which lost all tightly binding peptides because of the Omicron mutations. At the same time, peptide presentation at the level of the whole virus was practically unaffected by the mutations. Given that we hypothesize that some individuals vaccinated with Spike protein-based vaccines could develop the impaired T-cell immune responses to the Omicron variant. Experimental verification of this hypothesis is warranted.

### Funding

The research was performed within the framework of the Basic Research Program at HSE University (Maxim Shkurnikov, Alexei Galatenko; study design, interpretation), the donation of SberBank to Faculty of Biology and Biotechnology at HSE University (Stepan Nersisyan, Alexander Tonevitsky; bioinformatics analysis, T-CoV portal development) and the Russian Science Foundation (Project No. 17-74-30019; Maria Zakharova, Irina Ishina, Inna Kurbatskaia, Azad Mamedov, Alexander Gabibov; HLA/peptide experiments). The funders had no role in study design, data collection and analysis, decision to publish, or preparation of the manuscript.

### Grant Disclosures

The following grant information was disclosed by the authors:
Basic Research Program at HSE University.
SberBank to Faculty of Biology and Biotechnology at HSE University.
Russian Science Foundation: 17-74-30019.

### Competing Interests

Stepan Nersisyan is an employee of Armenian Bioinformatics Institute (ABI). Alexander Tonevitsky is an employee of Art Photonics GmbH.

### Author Contributions

- Stepan Nersisyan conceived and designed the experiments, performed the experiments, analyzed the data, prepared figures and/or tables, authored or reviewed drafts of the paper, and approved the final draft.
- Anton Zhiyanov performed the experiments, authored or reviewed drafts of the paper, and approved the final draft.
- Maria Zakharova performed the experiments, authored or reviewed drafts of the paper, and approved the final draft.

- Irina Ishina performed the experiments, authored or reviewed drafts of the paper, and approved the final draft.
- Inna Kurbatskaia performed the experiments, authored or reviewed drafts of the paper, and approved the final draft.
- Azad Mamedov performed the experiments, authored or reviewed drafts of the paper, and approved the final draft.
- Alexei Galatenko conceived and designed the experiments, analyzed the data, authored or reviewed drafts of the paper, and approved the final draft.
- Maxim Shkurnikov conceived and designed the experiments, analyzed the data, authored or reviewed drafts of the paper, and approved the final draft.
- Alexander Gabibov conceived and designed the experiments, analyzed the data, authored or reviewed drafts of the paper, and approved the final draft.
- Alexander Tonevitsky conceived and designed the experiments, analyzed the data, authored or reviewed drafts of the paper, and approved the final draft.

## Data Availability

All source codes are available at GitHub: https://github.com/s-a-nersisyan/TCellCovid19Atlas.

## Supplemental Information

Supplemental information for this article can be found online at http://dx.doi.org/10.7717/peerj.13354#supplemental-information.

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
