# Peer review of "Alterations in SARS-CoV-2 Omicron and Delta peptides presentation by HLA molecules"

_PeerJ, doi:10.7717/peerj.13354_

## Round 0.1 · original submission · Minor Revisions

Please address concerns of all reviewers and amend your manuscript accordingly.

Reviewer 1 ·

Basic reporting

Stepan Nersisyan and collaborators offer an interesting evaluation of alterations in SARS-CoV-2 Omicron and Delta peptides presentation by HLA molecules. The manuscript looks professionally written, excepting some fine-minor mistakes in grammar and typos.
In the introduction section, the authors just mention HLA class II viral-antigenic presentation; I recommend reinforcing this background since viral antigens are presented in the HLA class I context.
The last paragraph in the introduction regarding two updates to the T-CoV portal is out of the section; please relocate to the results section and avoid duplicating this information.
Figures look well and are highly informative. Please consider being more descriptive for titles for supplementary materials or include a legend describing their contents.
Although several pieces of research have described association studies related to genetic susceptibility with the HLA system, results vary between populations. Also, some alleles have been associated with protection in other viral respiratory diseases, particularly interesting those associated with class I alleles (example: doi: 10.1155/2018/3174868); please comment.

Experimental design

In my opinion, the submission defines properly the research question, which is relevant for the scientific forum.
Regarding the methods employed, the authors should expand the description for HLA-DRB1*03:01/peptide binding experiments section, beyond the ELISA standard protocols. Including trademarks/distributors and country.
Line 240 says: we then manually adjusted the results for HLA-DPA1/DPB1 and HLA-DQA1/DQB1 links. Please explain briefly in the text this adjustment.

Validity of the findings

Definitively, the higher strength is the validation employing peptide binding experiments. The authors compared peptide presentation profiles for the Omicron, Delta, and Wuhan SARS-CoV-2 variants in this manuscript. Only HLA-B*35:03 had one more Omicron/Delta tight binding peptide in NSP12 protein. Also, they predicted the complete absence of T-cell immunity evasion at the level of the whole virus.
When the analysis was limited to the Spike protein, eight alleles showed the enhanced ability of Omicron/Delta peptide presentation. In comparison, the four remaining alleles showed a significant decrease in the number of tight binders.
Data analysis apparently includes all information, no repository is mentioned, also, two updates to the T-CoV portal are described.
In a very recent paper, Zhang Y (doi.org/10.1038/s41431-022-01070-5) provides new insight into allelic influence on the HLA expression in association with SARS-CoV-2 and other common viral infectious diseases. Please discuss these findings in the context of your results.
Also, Xiao C (doi: 10.1016/j.isci.2022.103934) shows that most of the predicted CD8+ T cell epitopes showed proper binding with HLA-A2, whereas epitopes from B.1.1.7 had lower binding capability than those from the ancestral strain. In addition, these peptides could effectively induce the activation and cytotoxicity of CD8+ T cells. Please analyze and discuss these findings regarding your work.
The same group (doi: 10.3389/fimmu.2021.764949) previously described that the variation of a dominant epitope will cause the deficiency of HLA-A*02:01 binding and T-cell activation, which subsequently requires the formation of a new CD8+ T-cell immune response in COVID-19 patients. Please analyze and discuss these findings regarding your work.

Additional comments

Minor comments:
Please include the official description for variants the first time are described: Delta (B.1.617.2) and Omicron (B.1.1.529), etc.
Line 195, correct Oсmicron by Omicron.
Line 226: and a whole HLA haplotype is inherited from each "patient"... did you mean "parent"? Please correct.

Reviewer 2 ·

Basic reporting

The authors compare viral peptide presentation for the three variants: Wuhan, Delta and Omicron, and the comparisons were separately conducted for the whole virus and Spike protein peptidomes. The authors employed T-CoV database to obtain the lists of viral peptides which were predicted to be tight binders (affinity 50 nM) for the worldwide prevalent HLA class I and class II alleles.

Experimental design

The experimental design by the authors is sound. The presented work is interesting and it provides useful information for researchers investigating the dynamical aspect of peptide presentation combining computational with experimental results.

Validity of the findings

The key finding of the authors are:
• absence of significant differences in the presentation of SARS-CoV-2-derived peptides by the most frequent HLA class I/II alleles and the corresponding HLA haplotypes
• destructive effect of the Omicron mutations on PINLVRDLPQGFSAL for HLA-DRB1*03:01 allele
• N211 deletion, L212I substitution and EPE 212-214 insertion dramatically declines the binding affinity of HLA-DRB1*03:01 and this peptide

Additional comments

The limitation of the study is that only set of peptides originating from the Spike proteins of the SARS-CoV-2 variants have been considered.

There are some points that should be addressed before the paper to be accepted:

1) The authors should describe briefly how the binding experiments were conducted, and not just referring to their previous work.
2) The author mention that not a single HLA class II allele passed 25% difference threshold when the analysis was performed at the level of the whole virus peptidome. It is not clear how did the authors choose hla-drb1*03:01 for further investigation?
3) Line 194: predictions obtained by netMHCIIpan, all corresponding “Oсmicron”. Correct this typo, and check thoroughly the paper.
4) Line 214: There was another HLA class II allele with differential peptide presentation. Which is the HLA class II allele? This is not clear.
5) In Figure 1 : -100% or -33% mean?

Annotated reviews are not available for download in order to protect the identity of reviewers who chose to remain anonymous.

Reviewer 3 ·

Basic reporting

Literature references were sufficient, the authors provided context and the article has structure. The conclusion section needs a rewriter and deleting the hypothesis.
Authors should be writing the aim in the introduction section.

Experimental design

The research show originality, bioinformatic analysis was rigorous, and peptide binding experiments had strong results.

Figure 2. Can you include the p-value in comparisons between Omicron and Delta Variants?

Figure 3. Can you include the p-value between HLA-DRB1*03:01+ and HLA-DRB1*03:01-

Line 171. “HLA-B*07:02, HLA-B*27:05, and HLA-A*32:01 had more than 50% difference in the number of presented Omicron/Delta peptides”; however, in figure 2, HLA-B*07:02 is the same value in both variants.

Figure 2. HLA-B*44:03 or HLA-B*08:01 show differences similar to HLA- A*32:01; do these alleles have biological importance? please discuss.

Validity of the findings

Line 315-317, authors add hypothesis; however, this is a conclusion section, please, delete the sentence and change to section discussion.

Additional comments

No comment

Annotated reviews are not available for download in order to protect the identity of reviewers who chose to remain anonymous.

---

## Round 0.2 · accepted · Accept

All issues pointed out by the reviewers were adequately addressed. Therefore your revised manuscript is acceptable in its present form.

Reviewer 2 ·

Basic reporting

The authors have improved the overall presentation of the paper, as suggested by the Reviewers.

Experimental design

The Material and Method section has been improved.

Validity of the findings

The results obtained from this work are sound and important particularly in the context of Covid-19 pandemic that we all are facing.

Additional comments

Accept in its current form

Reviewer 3 ·

Basic reporting

The introduction shows context and structure.
References are sufficient.
The figures are well-described.

Experimental design

The results are novelty and robust.

Validity of the findings

Methods are described with sufficient detail.

Additional comments

The authors heeded the observations made previously.